# Deciphering High-Temperature-Induced Lignin Biosynthesis in Wheat through Comprehensive Transcriptome Analysis

**DOI:** 10.3390/plants13131832

**Published:** 2024-07-03

**Authors:** Junjie Han, Zhenlong Wang, Xianghu Wu, Jianqiang Xia, Lihong Wang, Zhong Wang, Yueqiang Zhang

**Affiliations:** 1Institute of Nuclear and Biological Technology, Xinjiang Academy of Agricultural Sciences, Urumqi 830091, China; hanjunjie_xjnky@163.com (J.H.); wangzhenlong@xaas.ac.cn (Z.W.); 17693138880@163.com (J.X.); lihongwang@xaas.ac.cn (L.W.); zhongwang@xaas.ac.cn (Z.W.); 2Graduate School (Office of Academic Construction), Shihezi University, Shihezi 832003, China; mooocha@163.com

**Keywords:** wheat, high temperature, lignin, PAL, L-phenylalanine

## Abstract

This study systematically investigated the physiological and molecular responses of the wheat mutant ‘XC-MU201’ under high-temperature stress through comprehensive transcriptome analysis and physiological measurements. RNA sequencing of 21 samples across seven different treatment groups revealed, through Weighted Gene Co-expression Network Analysis (WGCNA), 13 modules among 9071 genes closely related to high-temperature treatments. Gene Ontology (GO) and Kyoto Encyclopedia of Genes and Genomes (KEGG) pathway analyses showed significant enrichment of lignin biosynthesis-related modules under high-temperature conditions, especially at the H-10DAT, H-20DAT, and H-30DAT time points. Experimental results demonstrated a significant increase in lignin content in high-temperature-treated samples, confirmed by tissue staining methods, indicating wheat’s adaptation to heat damage through lignin accumulation. The phenylalanine ammonia-lyase gene (TaPAL33) was significantly upregulated under high-temperature stress, peaking at H-30DAT, suggesting its critical role in cellular defense mechanisms. Overexpression of TaPAL33 in the wheat variety ‘Xinchun 11’ enhanced lignin synthesis but inhibited growth. Subcellular localization of GFP-labeled TaPAL33 in tobacco cells showed its distribution mainly in the cytoplasm and cell membrane. Transgenic wheat exhibited higher PAL enzyme activity, enhanced antioxidant defense, and reduced oxidative damage under high-temperature stress, outperforming wild-type wheat. These results highlight TaPAL33’s key role in improving wheat heat tolerance and provide a genetic foundation for future research and applications.

## 1. Introduction

Wheat (*Triticum aestivum* L.), a fundamental crop for global food security, serves as a crucial dietary staple and economic foundation for billions of people worldwide. Predominantly cultivated in temperate regions, wheat is highly sensitive to elevated temperatures, which compromise its physiological and biochemical integrity, thus hindering its metabolic functions [1]. These thermal stresses lead to reduced growth rates and lower yields, significantly affecting commercial productivity [2]. Addressing these challenges necessitates the development and implementation of robust adaptive strategies to protect wheat from sudden and extreme temperature fluctuations.

Recent research has increasingly focused on enhancing thermal tolerance in wheat through genetic interventions, particularly targeting pathways like lignin biosynthesis—a critical phenolic biopolymer that ensures structural integrity in plants [3]. Lignin plays a pivotal role in cell differentiation by consolidating the secondary cell wall. It replaces the aqueous phase and synergistically integrates with cellulose and other polysaccharides, augmenting mechanical strength and establishing an effective hydrophobic barrier [4]. This structural reinforcement is essential in vascular tissues, where it mitigates the adverse impacts of negative transpirational pressure. Notably, lignin deficiencies can lead to catastrophic failures in these tissues, jeopardizing plant stability and function [5].

Lignin biosynthesis is predominantly mediated through the phenylpropanoid pathway, utilizing L-phenylalanine (L-Phe) as the initial precursor. The enzyme phenylalanine ammonia-lyase (PAL) plays a crucial role in this pathway by catalyzing the conversion of L-Phe to cinnamic acid, thereby facilitating the onset of lignin polymerization [6]. Additionally, recent studies have identified an alternative, more energy-efficient pathway for lignin biosynthesis in cereals. This pathway involves tyrosine ammonia-lyase (TAL), which incorporates L-tyrosine (L-Tyr) into lignin biosynthesis, demonstrating a significant adaptation in the metabolic routes of these plants [7]. Under high-temperature stress conditions, there is a discernible upregulation of phenylalanine ammonia-lyase (PAL) and other genes associated with lignin biosynthesis, suggesting that enhancing these pathways may improve wheat’s thermal tolerance [8]. This research examines this hypothesis through experimental trials using the wheat mutant XC-MU201, cultivated under various high-temperature regimes in comparison to control conditions. Utilizing RNA-seq analysis of 21 samples, complemented by integrative data analysis from Gene Ontology (GO), Kyoto Encyclopedia of Genes and Genomes (KEGG), Weighted Gene Co-expression Network Analysis (WGCNA), and qRT-PCR, the study aims to elucidate alterations in the lignin synthesis pathway and their consequent effects on wheat’s adaptive responses to heat stress.

This research aims to achieve several objectives: First, it seeks to elucidate the molecular mechanisms enabling wheat to withstand high-temperature stress through genetic modifications, with a particular emphasis on lignin metabolism. Second, the study investigates the differential expression of genes involved in these adaptive mechanisms. Third, it evaluates the physiological traits of transgenic wheat lines that overexpress critical genes, such as *TaPAL33*. Ultimately, this investigation aims to provide foundational insights to support the development of heat-resistant wheat varieties. These advances are essential for enhancing agricultural resilience in the face of increasing climate variability.

## 2. Results

### 2.1. Phenotypic and Physiological Responses to High-Temperature Stress

To assess the response of wheat to elevated temperature stress, plants in the three-leaf stage were exposed to 35 °C for periods of 10, 20, and 30 days. This thermal stress notably impaired typical growth patterns, evident from symptoms such as leaf wilting, curling, and diminished water content (Figure 1A,B). Subsequent analyses indicated physiological adjustments; plants subjected to prolonged high temperatures demonstrated significant increases in malondialdehyde (MDA), proline (Pro), and soluble sugars compared to controls maintained under ambient conditions (Figure 1C–E). These physiological changes imply that wheat activates defensive strategies to counteract oxidative stress, regulate osmotic balance, and enhance energy reserves, thereby improving thermal resilience. Furthermore, the activities of antioxidant enzymes such as superoxide dismutase (SOD) and catalase (CAT) were substantially elevated (Figure 1F,G), playing crucial roles in detoxifying reactive oxygen species (ROS) and minimizing cellular damage. Conversely, peroxidase (POD) activity showed no significant alteration, registering only a slight increase (Figure 1H).

### 2.2. Transcriptome Dynamics in Wheat Exposed to High-Temperature Stress

To investigate differential gene expression in wheat subjected to both ambient and elevated temperature conditions, comprehensive RNA sequencing was performed. This analysis yielded 968,325,976 raw reads, which were processed into 959,033,928 clean reads, totaling 125.88 Gb of high-quality clean bases from 21 samples. The sequencing quality was good, with Q30 base percentages ranging from 91.65% to 93.01% and an overall average GC content of 51.60%, as detailed in Appendix A. Following de novo transcriptome assembly, the clean reads were aligned to the IWGSC RefSeq v2.1 wheat reference genome. Mapping efficiencies varied from 49.93% to 85.07%, indicating comprehensive coverage and alignment, as detailed in Appendix A. Pearson correlation coefficients were used to determine correlations between samples, revealing inter-sample relationships. A cluster plot, derived from these data, illustrates the transcriptomic correlations under different treatments, as shown in Appendix A, confirming the robust repeatability of the sequencing data across the 21 samples. Principal Component Analysis (PCA) was employed to explore similarities among the samples. By applying dimensionality reduction to the first and second principal components (PC1 and PC2), PCA efficiently revealed similarities across sample replicates, as illustrated in Appendix A.

Differentially expressed genes (DEGs) were identified from the 21 samples using a fold-change threshold of ≥2 and a strict false discovery rate (FDR) of <0.05. Gene Ontology (GO) enrichment analysis was conducted using DAVID bioinformatics platform (v2022q4). After annotation, the transcripts were compared against the online Kyoto Encyclopedia of Genes and Genomes (KEGG) database and subsequently mapped to KEGG pathways. In the comparison groups H-10DAT_vs_N-0DAT, H-20DAT_vs_N-20DAT, and H-30DAT_vs_N-30DAT, the most prominently identified KEGG pathways were “Protein Processing in Endoplasmic Reticulum” and “Carbon Metabolism” related to photosynthesis. Additional pathways such as “Plant Hormone Signal Transduction”, “Glycerolipid Metabolism”, and “Metabolic Pathways” were also observed (Figure 2A,C,D). Specifically, in comparison group H-10DAT_vs_N-10DAT, the most prevalent KEGG categories were “Protein Processing in the Endoplasmic Reticulum” and “Photosynthesis”, followed by “Biosynthesis of Amino Acids” and “Carbon Metabolism” (Figure 2B). Notably, pathways associated with lignin metabolism, such as “Phenylalanine Metabolism” and “Biosynthesis of Phenylalanine, Tyrosine, and Tryptophan”, were consistently observed across the four groups, underscoring their importance. Additionally, significant enrichment in ‘Pyruvate Metabolism’ and ‘Peroxisome’ pathways was observed, with the former providing precursor molecules for the phenylpropanoid pathway and the latter facilitating lignin accumulation. In the comparative analysis of DEGs among the four groups, a total of 2056 DEGs were identified (Figure 2E). Remarkably, group H-10DAT_vs_N-0DAT exhibited the highest number of DEGs, indicating a rapid gene response to high-temperature stress and suggesting a potential gradual adaptation to elevated temperatures in later stages.

### 2.3. Construction of Co-Expression Networks under High-Temperature Conditions Reveals Links to Lignin Biosynthesis

In this study, transcriptome data from various treatment periods of ‘XC-MU201’ were meticulously analyzed. Low-abundance genes were initially excluded, and the resulting dataset, comprising 9071 genes, was processed using Weighted Gene Co-expression Network Analysis (WGCNA) software package (Version: WGCNA 1.72-5; Appendix A). This analysis segmented the transcriptomic data from each treatment period into distinct co-expression modules via WGCNA. Within the WGCNA framework, each module was represented as a branch on a dendrogram, with individual leaves symbolizing genes, as illustrated in the hierarchical clustering tree (Figure 3A). These dendrogram branches were methodically pruned to define discrete modules. Each module was then characterized by its correlation with the plant’s response to high-temperature treatments. WGCNA identified thirteen unique modules, each distinguished by a specific color (Figure 3B, Appendix A). The correlation of each module with the target genes—whether positive or negative, and regardless of magnitude—highlights their significant association with the biological processes active during the respective periods of analysis.

The correlation analysis of WGCNA modules with traits revealed significant variability in gene expression under high-temperature conditions, demonstrating both positively and negatively correlated genes within these modules. This study identified six modules that showed strong correlations to high-temperature treatments, as illustrated in Figure 3B: MElightyellow in N-0DAT (*r* = 0.97, *p* = 2 × 10^−4^); MEgray60 in H-10DAT (*r* = 0.89, *p* = 0.007); MEgreenyellow in H-20DAT (*r* = 0.96, *p* = 5 × 10^−4^); MEred in H-30DAT (*r* = 0.91, *p* = 0.004); MEblack in N-30DAT (*r* = 0.81, *p* = 0.03); and MEbrown in N-30DAT (*r* = 0.84, *p* = 0.02). Further analyses explored the functional categories of genes within these modules using GO, KEGG, and feature gene expression profiling techniques. In the MElightyellow module, the top GO annotations were “chromosome organization” for biological processes, “protein-DNA complex” for cellular components, and “DNA binding” for molecular functions, as detailed in Appendix A. Additional categories significantly enriched included “protein folding”, “chromosome”, and “protein heterodimerization activity”. KEGG pathway analysis underscored “protein processing in the endoplasmic reticulum” and “DNA replication” as notably enriched pathways (Appendix A). These results suggest that gene expression under standard conditions predominantly reflects essential biological processes such as chromosomal dynamics within the nucleus, protein–DNA interactions, and gene expression regulation. In the MEblack module, the highest GO enrichment was observed in “DNA packaging complex” and “nucleosome” (Appendix A), while the KEGG pathway analysis highlighted “metabolic pathways” and “protein processing in the endoplasmic reticulum” as significantly enriched (Appendix A). Conversely, the MEbrown module exhibited a unique pattern, significantly enriching terms associated with photosynthesis, such as “photosystem”, “response to light stimuli”, “photosynthesis”, and “chlorophyll binding” (Appendix A). KEGG pathways in this module predominantly focused on “metabolic pathways”, “biosynthesis of secondary metabolites”, and “carbon metabolism” (Appendix A). Additionally, the expression profiles of the MEblack and MEbrown modules were most pronounced in N-30DAT (Appendix A), possibly reflecting the wheat’s adaptive mechanisms to accumulate substantial nutrients and energy during this period to support reproductive growth.

This study uncovered distinct patterns of gene expression in lignin biosynthesis across several modules, each corresponding to different high-temperature treatment durations. Specifically, the MEgrey60 module, associated with the H-10DAT treatment, comprises nine genes involved in the “lignin biological process” (GO: 0009809), with six genes upregulated and three downregulated (Figure 4A). Additionally, KEGG annotations for this module indicate enrichment in pathways critical to lignin biosynthesis, namely “Phenylpropanoid biosynthesis” and “Phenylalanine metabolism” (Figure 4B). Furthermore, the MEgreenyellow module, for H-20DAT, includes 11 genes associated with the “lignin biological process”, with eight genes upregulated and three downregulated (Figure 4C). The KEGG annotations highlight enrichment in “Phenylalanine metabolism”, “Phenylalanine, tyrosine, and tryptophan biosynthesis”, and “Phenylpropanoid biosynthesis” pathways (Figure 4D). Similarly, the MEred module, corresponding to H-30DAT, shows enrichment in the “lignin biosynthetic process” (seven genes upregulated and two downregulated) and the “L-phenylalanine catalytic process” (GO: 0006559) (six genes upregulated and three downregulated) (Figure 4E). The KEGG annotations also reveal enrichment in “Phenylalanine metabolism”, “Phenylpropanoid biosynthesis”, and “Phenylalanine, tyrosine, and tryptophan biosynthesis” pathways (Figure 4F). Across these modules, which are linked to different durations of high-temperature treatment, genes involved in lignin synthesis generally exhibit higher expression levels compared to those in the control group (Appendix A). These findings highlight the critical regulatory role that lignin synthesis pathways play in plant responses to elevated temperatures and stress, underscoring their potential as targets for enhancing stress resilience.

### 2.4. Lignin Synthesis and Related Gene Expression in Response to High-Temperature Stress

This study involved collecting wheat leaf samples under various treatment conditions and subsequently measuring their lignin content after a drying process (Figure 5). Initially, at the experiment’s onset (0DAT), the lignin levels were relatively low across both standard and high-temperature conditions. However, as the exposure duration increased, a consistent upward trend in lignin content was observed in all experimental groups. Notably, significant increases in lignin content were recorded on the 10th (10DAT) and 20th days (20DAT) in the high-temperature treatment group compared to the control group, which was maintained at room temperature. This pronounced enhancement suggests that elevated temperatures substantially accelerate lignin accumulation. Under continued high-temperature conditions, lignin levels eventually stabilized, indicating that the increase in lignin content, induced by prolonged high temperatures, may serve as an adaptive mechanism enabling wheat plants to mitigate thermal stress.

To elucidate the effects of high temperatures on lignin accumulation in wheat leaves, this study utilized a safranin staining solution during the N-20DAT, H-20DAT, N-30DAT, and H-30DAT stages. This staining method effectively highlights lignin by binding to this complex polymer, providing a vivid and detailed visualization of its distribution within plant tissues. Results demonstrated that under high-temperature conditions, there was a significant increase in lignin content within the leaves, which manifested as a more intense staining effect (Figure 6). Notably, the cell walls at the H-20DAT and H-30DAT stages displayed deeper red staining, indicative of substantial lignin enrichment in these cells (Figure 6B,D, marked by arrows). This observation supports the hypothesis that elevated temperatures may enhance the biosynthesis or stability of lignin within plant cell walls, serving as a protective mechanism against thermal stress. Conversely, the leaves from the N-20DAT and N-30DAT stages exhibited lower lignin concentrations and a more scattered distribution (Figure 6A,C), possibly due to fewer stress-induced biochemical pathways being activated. The contrasting staining patterns between leaves subjected to high- and room temperature conditions not only highlight the significant impact of environmental stress on lignin biosynthesis but also suggest a crucial role for lignin in enhancing plant resilience to high-temperature stress [9,10]. Further investigations into the gene expression profiles associated with lignin production during these stages could provide deeper insights into the underlying mechanisms facilitating these adaptations.

This study investigates the regulatory mechanisms governing lignin biosynthesis in wheat under high-temperature stress and elucidates its potential role in responding to such conditions. Lignin biosynthesis predominantly occurs via the phenylpropanoid pathway [11], involving several critical enzymes, including phenylalanine ammonia-lyase (PAL) and tyrosine ammonia-lyase (TAL). These enzymes catalyze the conversion of phenylalanine (Phe) and tyrosine (Tyr) into cinnamic acid and p-coumaric acid, respectively (Figure 7). Subsequent hydroxylation, methylation, and reduction reactions transform these acids into lignin monomeric precursors, with key enzymes involved being 4-coumaric acid CoA ligase (4CL), cinnamyl alcohol dehydrogenase (CAD), and caffeic acid O-methyltransferase (COMT). Transcriptomic analysis of wheat samples subjected to various high-temperature treatments revealed significant differential expression of genes related to lignin synthesis, such as *PAL* and *C4H* (cinnamate 4-hydroxylase) (Figure 7, Appendix A). Among the 37 PAL genes and 41 *CAD* genes identified, 8 *PALs* and 6 *CADs* exhibited significantly higher expression levels under high-temperature conditions. Additionally, other genes in the phenylpropanoid pathway, such as *4CL* (one out of six), *CAD* (6 out of 41), *COMT* (6 out of 30), *F5H* (ferulate 5-hydroxylase, one out of seven), *CCoAOMT* (caffeoyl CoA O-methyltransferase, 4 out of 19), and *CHS* (chalcone synthase, 3 out of 13), also showed similar expression patterns. These genes collectively contribute to the synthesis of various lignin monomers, including p-hydroxyphenyl (H-type), guaiacyl (G-type), and syringyl (S-type) lignin. Interestingly, the expression levels of *PAL* genes during the high-temperature treatment periods (H-10DAT, H-20DAT, and H-30DAT) were elevated compared to the control group, suggesting that *PALs* play a crucial role in wheat’s adaptive response to high temperatures. Similarly, the *CAD* enzyme, which primarily catalyzes the conversion of cinnamaldehyde to cinnamyl alcohol—the final step in lignin monomer synthesis—also exhibited higher expression levels at specific high-temperature treatment stages compared to the control. These findings indicate that high temperatures significantly impact the activity of the phenylpropanoid pathway in wheat, particularly affecting the expression of key genes involved in lignin synthesis. Consequently, these changes may represent a defensive mechanism by which wheat copes with high-temperature stress.

### 2.5. Structural Insights and Conservation Patterns of PAL Genes

This study investigates the primary function of phenylalanine ammonia-lyase (PAL), which is to catalyze the deamination of phenylalanine (Phe), producing trans-cinnamic acid and ammonia. This reaction is crucial for bridging primary and secondary metabolites, pivotal for plant stress resistance, growth regulation, and defense mechanisms. Through RNA sequencing, 37 *PAL* genes were identified, designated as *TaPAL1* through *TaPAL37*.

To comprehensively characterize the structural diversity and protein conservation within the wheat *TaPAL* gene family, this study employed Gene Structure Display Server (GSDS) platform (http://gsds.cbi.pku.edu.cn/ (accessed on 20 February 2024)) to analyze the exon–intron structures of each gene. Additionally, MEME suite (https://meme-suite.org/meme/tools/meme (accessed on 20 February 2024)) was used to explore the sequence motif diversity of the TaPAL proteins. The analysis identified ten distinct motifs in wheat, revealing significant variations in the number and types of motifs across different TaPAL proteins. These variations underscore the non-conservation of amino acid sequences and gene structures, as well as functional diversity (Figure 8A). For instance, *TaPAL3* (*TraesCS1A02G094900*) lacks motifs six, eight, and ten; *TaPAL13* (*TraesCS2A02G196400*) and *TaPAL19* (*TraesCS2B02G224300*) are missing motifs six and ten, respectively; both *TaPAL30* (*TraesCS5B02G468400*) and *TaPAL33* (*TraesCS6B02G258500*) are deficient in two motifs; whereas *TaPAL32*, *TaPAL35*, and *TaPAL36* share a similar structure, absent of motif seven. Notably, *TaPAL21* contains only motif six, which includes the catalytic active domain “Lyase-aromatic”, due to its shorter exon. Further analyses using TBtools of the cis-acting elements in the promoter regions of 37 *TaPAL* genes identified a total of 464 hormone-responsive elements, encompassing auxins (27), gibberellins (30), salicylic acid (14), abscisic acid (163), and methyl jasmonate (MeJA) responses (230) (Figure 8B, Appendix A). Photoresponse-related elements were the most prevalent, numbering 380. Additional elements associated with stress responses include defense and stress responsiveness (5), low temperature (44), anaerobic induction (73), and drought induction (36), as well as elements related to cell cycle regulation, circadian rhythm control, meristem expression, and differentiation of palisade mesophyll cells. These findings not only highlight the critical role of *PAL* genes in lignin synthesis but also suggest their involvement in hormone regulation and stress response.

Moreover, the exon and intron structures of the *TaPAL* gene family were analyzed (Figure 8C). Most members possess two to three exons; however, *TaPAL3*, *9*, *12*, *13*, *21*, *24*, *31*, *32*, *34*, and *35* have only one exon region. These structural analyses further emphasize the complexity and multifunctionality of the *TaPAL* gene family in wheat.

Phylogenetic analysis is a reliable method for elucidating the functions of wheat TaPAL proteins by examining their genetic history and sequence consistency among PAL members. This study constructed a phylogenetic tree comprising 37 wheat TaPALs and 16 *Arabidopsis* full-length PAL protein sequences (Figure 9). The topological structure of the tree was analyzed using the Maximum Likelihood (ML) method. Based on the existing classification of the PAL family in *Arabidopsis* [12], five distinct groups were identified: Groups I, II, III, IV, and V. Group V contained the largest number of genes, with 24 members. Subsequently, Group II, consisting solely of TaPALs, comprised 11 genes and formed a unique clade. In contrast, the genes in Group III originated exclusively from *Arabidopsis*. These findings highlight the significant evolutionary divergences between the *PAL* genes of wheat and those of *Arabidopsis*.

### 2.6. Validation of RNA-Seq Analysis Using qRT-PCR

To verify the accuracy of the RNA-seq results, qRT-PCR was conducted to measure the transcription levels of differentially expressed genes in eight *TaPALs* identified in this study. Both RNA-seq and qRT-PCR analyses exhibited consistent trends for these eight DEGs, thereby substantiating the transcriptome analysis results depicted in Figure 10. The expression profiles revealed that, compared to the control environment, the expression levels of most genes in the *PAL* pathway were elevated in the high-temperature environment at least at one developmental stage. This suggests a significant thermoregulatory effect on *TaPAL* gene expression, which likely influences lignin biosynthesis at elevated temperatures. These findings provide a foundation for further investigation into the role of *TaPAL* genes under thermal stress conditions. Details of the gene-specific primers used are provided in Appendix A.

### 2.7. Overexpression of TaPAL33 Boosts Lignin Levels and Mitigates Heat Stress in Wheat via Enzymatic Activity

Transcriptomic investigations have revealed a wealth of wheat genes that respond to thermal stress, particularly those involved in cell wall development, providing pivotal knowledge for enhancing wheat’s resilience to elevated temperatures. For detailed functional analysis, phenylalanine ammonia-lyase (*PAL33*, *TraesCS6B02G2258500*), a key enzyme in lignin biosynthesis, was selected. Data indicated that the expression of *TaPAL33* peaked at H-30DAT under various treatment regimens (Figure 10).

Furthermore, *TaPAL33* was overexpressed in wheat cultivar “Xinchun 11”, cultivated in Xinjiang, resulting in the generation of more than 30 independent transgenic lines. Phenotypic evaluation of two transgenic lines, OE1 and OE2, which exhibited the highest levels of *TaPAL33* expression, revealed significant growth inhibition (Figure 11A). This suggests a potential role for *TaPAL33* in regulating cellular elongation. Further exploration of the protein’s cellular localization and function involved the fusion of its coding sequence with a GFP tag and subsequent transformation into tobacco leaves. Microscopic analysis showed that the TaPAL33–GFP fusion protein was predominantly localized in the cytoplasm and at the cell membrane (Figure 11B). Additionally, transgenic (OE) and wild-type (WT) lines underwent a 30-day treatment under both standard and high-temperature conditions. Subsequent physiological assessments demonstrated that even under normal conditions, the lignin content in OE lines exceeded that of the controls. With high-temperature exposure, lignin levels increased significantly across all lines, with the most pronounced increases observed in the OE lines (*p* < 0.01). PAL enzyme activity was significantly enhanced in the OE lines. After thermal stress, flavonoid concentrations in the leaves of OE lines were substantially higher than those in WT lines (*p* < 0.01), although differences in anthocyanin content between the lines were not statistically significant (*p* > 0.05). Moreover, overexpression of *TaPAL33* significantly elevated the activities of antioxidant enzymes (SOD, POD, and CAT) and reduced the levels of malondialdehyde (MDA) in the leaves (Figure 11C). In summary, *TaPAL33* plays a critical regulatory role in wheat development and thermotolerance by modulating lignin metabolism within the cell wall. These findings provide valuable insights into strategies to enhance wheat adaptability to high-temperature environments, underscoring the potential genetic targets for crop improvement.

## 3. Discussion

Our study provides comprehensive insights into the physiological and molecular responses of wheat to high-temperature stress, with a particular focus on lignin biosynthesis. Through WGCNA of RNA-seq data, we identified 13 gene clusters associated with high-temperature and control conditions. Notably, differential genes related to lignin synthesis were enriched in clusters such as MElightyellow at N-0DAT, MEgray60 at H-10DAT, MEgreenyellow at H-20DAT, MEred at H-30DAT, and MEblack and MEbrown at N-30DAT. Lignin, an essential secondary metabolite, primarily originates from the phenylpropanoid metabolic pathway in plant cells [13]. GO and KEGG pathway analyses of the high-temperature-related modules (MEgray60, MEgreenyellow, and MEred) revealed significant enrichment in the “lignin biosynthesis process” and “phenylpropanoid biosynthesis” pathways (Figure 4). Specifically, the MEgreenyellow module displayed a higher number of differentially expressed genes under high-temperature conditions, suggesting an increase in lignin synthesis-related gene activity (Figure 4C,D). This indicates that high temperatures indeed induce changes in the lignin synthesis pathway in wheat.

To corroborate these findings, we measured lignin content in wheat leaves across various treatment periods (Figure 5). High temperatures significantly increased lignin accumulation, which continued to rise with prolonged exposure. Furthermore, safranin staining of leaves under high-temperature and control conditions revealed substantially greater areas of lignin enrichment under high temperatures, consistent with the measured lignin content (Figure 6).

High temperatures alter gene expression and plant metabolism, affecting numerous biological functions. Lignin deposition strengthens the cell wall, enabling plants to resist damage caused by high-temperature stress [14]. The pathways and expression levels of genes related to lignin synthesis in wheat under high-temperature and control conditions were mapped (Figure 7). The biosynthesis of lignin monomers involves a series of enzymatic reactions catalyzed by at least 17 enzymes. *PAL* and *C4H*, genes of the phenylpropanoid pathway, mark the initiation of lignin synthesis. Differential expression of eight *PALs* and two *C4Hs* under high temperatures suggests an early impact of heat on lignin synthesis. PAL, a critical enzyme, catalyzes the conversion of substrates into cinnamic acid (or *p*-coumaric acid) at the initial stage of lignin precursor biosynthesis. Research indicates that reducing PAL and C4H activity in tobacco through antisense inhibition leads to decreased lignin content and altered subunit composition, demonstrating the importance of these enzymes in plant growth and development [15,16]. Downregulation of *PAL*, *C4H*, *4CL*, and *C3H* significantly impacts lignin content [17,18,19], reinforcing the structural integrity of the plant vascular system and facilitating efficient water transport through enhanced hydrophobic cross-linking between cell wall polysaccharides and lignin [20]. This cross-linking also provides structural support and resistance to cell collapse under the tension of water transport [21].

The 4-coumaric acid CoA ligase (4CL) plays a pivotal role as a key enzyme in the lignin biosynthesis pathway, while cinnamyl CoA reductase (CCR) and cinnamyl alcohol dehydrogenase (CAD) are responsible for synthesizing H-type lignin. Our study indicates that high-temperature treatments lead to increased expression levels of *4CL* and *CCR*. Additionally, the synthesis of H-type lignin is influenced by enzymes such as caffeic acid O-methyltransferase (COMT), caffeoyl-CoA O-methyltransferase (CCoAOMT), and ferulate 5-hydroxylase (F5H), which further contribute to the formation of G-type and S-type lignin. The expression levels of *COMT* and *CCoAOMT* exhibited an upward trend in the later stages of high-temperature exposure, while *F5H* expression was upregulated during the early stages (Figure 7). These changes in gene expression ultimately led to an increase in lignin content under high-temperature conditions, suggesting that modifications in lignin synthesis genes serve as an emergency defense mechanism against high temperatures, thus enhancing the survival of wheat.

Lignin is essential for the biosynthesis of plant cell walls, particularly under heat stress conditions, where it exerts a significant protective effect. According to our findings, the synthesis of lignin can be enhanced by upregulating the *TaPAL33* gene, enabling plant cell walls to maintain their structural and functional integrity under elevated temperatures. Increased lignin content not only enhances the rigidity and hydrophobicity of the cell wall, effectively reducing water loss induced by high temperatures, but also diminishes the penetration of oxidative stress agents by decreasing cell wall porosity. Furthermore, the augmentation of antioxidant enzyme activities, such as superoxide dismutase (SOD), peroxidase (POD), and catalase (CAT), reduces the formation of reactive oxygen species (ROS) [22,23]. The synthesis of lignin also synergizes with the biosynthetic pathways of flavonoids, which respond to other environmental pressures. Under high-temperature conditions, flavonoid content significantly increased in plants overexpressing *TaPAL33* (Figure 11C). This synergistic interaction not only boosts the plant’s tolerance to high temperatures but also enhances its protection against UV-B radiation [24]. Therefore, the co-regulation of lignin and flavonoids embodies a comprehensive defense strategy, enabling plants to cope with multiple environmental pressures. However, the increased lignin content involves certain ecological and physiological trade-offs, such as potential inhibition of cell elongation, which could adversely affect plant growth and development and may even impact reproductive success [25]. This phenomenon was observed in our study where two transgenic lines, OE1 and OE2, overexpressing *TaPAL33* exhibited partial growth inhibition (Figure 11A).

In the context of global climate change and increasing extreme temperatures, enhancing crop heat tolerance is of paramount importance and has profound global implications. Future research should investigate genetically modified approaches that combine heat tolerance with other desirable agronomic traits and assess the ecological and sustainability impacts of these genetically modified crops through long-term field studies [26]. Additionally, exploring the natural variation in lignin content among different varieties will provide crucial insights for designing more effective breeding strategies.

## 4. Materials and Methods

### 4.1. Wheat Growth Conditions and High-Temperature Treatments

#### 4.1.1. Development of Heat-Resistant Mutant Wheat

The XC-MU201 mutant is a high-yield and heat-resistant wheat line developed by the Wheat Crop Research Institute at Shihezi University’s School of Agriculture through mutagenesis of the ‘Xinchun 11’ variety using ethyl methanesulfonate (EMS). This mutant was chosen for its known heat tolerance, which enables researchers to focus on specific pathways activated under heat stress conditions. By comparing it with wild-type wheat, the study aims to identify key differences in genetic response and physiological adaptation. Preliminary studies have shown that XC-MU201 exhibits enhanced lignin biosynthesis under heat stress, making it an ideal candidate for this research.

#### 4.1.2. Designing High-Temperature Stress Experiments for Wheat

Uniform-sized wheat seeds were disinfected with 10% (*v*/*v*) hydrogen peroxide for 20 min, followed by several rinses with distilled water. The seeds were then placed in dark culture dishes to germinate for two days. After germination, the seedlings were transferred to flowerpots containing sterile sand and incubated under controlled environmental conditions: 10 °C during a 9 h day and 7 °C throughout a 15 h night, with 60% relative humidity and a photon flux density of 500 µmol m^−2^ s^−1^. The seedlings were irrigated with Hoagland nutrient solution. One week later, the conditions were adjusted to 25 °C and 55% relative humidity. At the three-leaf stage, the seedlings were divided into two groups using a randomized block design. The first group was maintained under these stable conditions for 0, 10, 20, and 30 days (N-0DAT, N-10DAT, N-20DAT, and N-30DAT, respectively). The second group was transferred to a preheating chamber where the temperature was gradually increased by 1 °C per hour to 35 °C. Following thermal acclimation, these seedlings were subjected to high-temperature stress of 35 °C during the day and 25 °C at night for durations of 10, 20, and 30 days (H-10DAT, H-20DAT, and H-30DAT, respectively). At each interval, the fully expanded third leaves were harvested, immediately flash-frozen in liquid nitrogen, and stored at −80 °C. Leaf samples for both sequencing and physiological analysis were collected simultaneously from identical positions on each plant to ensure consistency. All experiments were repeated at least three times under identical light cycles to ensure the reliability of the results.

### 4.2. Measurement of Physiological Parameters

To quantify total dissolved sugar content in seedling tissues, the anthrone colorimetric method was employed, following the protocol described by Al-Sheikh et al. [27]. Standard curves were prepared using glucose as a calibration standard (Sigma-Aldrich, Shanghai, China). The procedure commenced with the mixing of 0.2 g of seedling tissue and 1 mL of distilled water, followed by heating at 100 °C for 10 min. Upon cooling, the mixture was centrifuged at 8000× *g* for 10 min and then diluted to a total volume of 10 mL. For the colorimetric analysis, a reaction mixture was prepared by combining 0.04 mL of sample extract, 0.04 mL of distilled water, 0.02 mL of anthrone reagent (1 g anthrone in 50 mL ethyl acetate, Vetec, Sigma-Aldrich, Shanghai, China), and 0.2 mL of 98% sulfuric acid (Vetec, Sigma-Aldrich, Shanghai, China). The mixture was subsequently heated at 95 °C for 10 min. Sugar concentrations were determined by measuring the absorbance at 620 nm using a Thermo Fisher Scientific scanning spectrophotometer (Waltham, MA, USA). Malondialdehyde (MDA) levels were assessed using the thiobarbituric acid reactive substances (TBARSs) method, as outlined by Marković et al. [28], involving absorbance measurements at 532 nm and 600 nm. The WST-8 assay, as detailed by Li et al. [29], was employed to quantify superoxide dismutase (SOD) activity, where WST-8 interacts with superoxide anions (O^2−^) in the presence of xanthine oxidase, forming a water-soluble formazan dye. The colorimetric assessment of the resultant products inversely quantified SOD activity. Peroxidase (POD) activity was quantified using the Amplex™ Red Hydrogen Peroxidase Assay Kit (A22188, Thermo Fisher Scientific, Waltham, MA, USA), based on the specific absorbance at 470 nm during H_2_O_2_ oxidation of substrates. Catalase (CAT) activity was measured using the protocol established by Li et al. [30] and the Amplex™ Red Catalase Assay Kit (A22180, Thermo Fisher Scientific). Proline content was determined using the proline content detection kit (BC0295, Beijing Solarbio Science & Technology Co., Ltd., Beijing, China) according to the sulfosalicylic acid (SA) extraction method. Approximately 200 mg of seedling tissue was homogenized in 2 mL of 3% SA (Vetec, Sigma-Aldrich, Shanghai, China), centrifuged at 13,000× *g* for 15 min at 4 °C, and mixed with 0.5 mL of acidic ninhydrin and acetic acid. The sample was heated at 100 °C for 30 min, cooled at 4 °C for an additional 30 min, and mixed with 1 mL of toluene (J&K Scientific, Beijing, China). The absorbance of the resulting 0.2 mL mixture was measured at 520 nm using a Thermo Scientific™ NanoDrop™ One spectrophotometer (Waltham, MA, USA).

### 4.3. Determination of Lignin Content

The determination of lignin concentration in leaves (mg/g dry weight) was conducted using a two-part experimental methodology. Leaf specimens were first dried at 50 °C for 24 h to achieve a uniform dry weight. Following the drying process, a precise 100 mg of the dried sample was subjected to an initial extraction with 1 mL of 12M H_2_SO_4_ (J&K Scientific, Beijing, China). Subsequently, 28 mL of distilled water was added to the extraction mixture, which was then maintained at a controlled temperature of 30 ± 0.5 °C for one hour with regular stirring to ensure uniformity in the reaction. The mixture was further hydrolyzed at 120 ± 5 °C for one hour to facilitate the breakdown of the sample’s lignin content. After hydrolysis, the mixture was quickly filtered to separate the dissolved lignin. The quantification of acid-soluble lignin in the filtrate was performed using spectrophotometry at a wavelength of 205 nm, following the quantitative analysis procedure described by Fredrik and Elisabeth [31]. This process was repeated three times to ensure the reliability of the results.

### 4.4. RNA-Seq and Bioinformatics Analysis

Appendix A provides a detailed description of the procedures for RNA extraction, meticulous assembly and sequencing of RNA-seq libraries, and comprehensive bioinformatic analysis. Reference genomic sequences and annotations were sourced from the Ensembl Plants database and subjected to stringent quality control (https://ftp.ensemblgenomes.ebi.ac.uk/pub/plants/release-56/ (accessed on 14 August 2023)). These controls included removing sequences of inferior quality and excluding short reads to ensure a dataset of high integrity for alignment against the gene annotations detailed in Appendix A. To identify differentially expressed genes (DEGs) across control and experimental conditions, the F-test within the Random Variance Model (RVM), optimized for small datasets, was utilized to enhance statistical validity. A rigorous analysis using a false discovery rate (FDR) meticulously identified DEGs with statistical significance. The expression levels of these DEGs were quantified using Cufflinks version 2.2.1 (http://cole-trapnell-lab.github.io/cufflinks/ (accessed on 25 December 2023)), employing the Fragments Per Kilobase of transcript per Million mapped reads (FPKM) metric for normalization, as detailed further in Trapnell et al. [32]. Gene Ontology (GO) terms for the DEGs were categorized using DAVID bioinformatics platform (v2022q4) (https://david.ncifcrf.gov/home.jsp (accessed on 25 December 2023)). The SMART tool (http://smart.embl-heidelberg.de/ (accessed on 12 January 2024)) played a critical role in identifying conserved structural domains within the *TaPAL* gene candidates, particularly those with intact coding sequences and the Lyase_aromatic domain. These genes were systematically renamed based on their chromosomal positions. Visualization of their structural configuration, specifically the exon–intron layout, was facilitated through the Gene Structure Display Server (GSDS) (http://gsds.cbi.pku.edu.cn/ (accessed on 12 January 2024)). Phylogenetic analysis was conducted using the Maximum Likelihood (ML) method via PhyML 3.0 (http://www.atgc-montpellier.fr/phyml/ (accessed on 16 January 2024)), which included 1000 bootstrap replications to verify the reliability of the evolutionary relationships depicted. This method also involved calculating evolutionary distances using the Poisson correction method and culminated in the visualization of the phylogenetic tree with iTOL v6 (https://itol.embl.de (accessed on 21 January 2024)), providing a comprehensive and clear depiction of the evolutionary lineage and intricate relationships among the studied taxa.

### 4.5. Histochemical Staining and Fluorescence Microscopy Detection

Lignified leaves were analyzed to determine lignin distribution using histochemical staining techniques and fluorescence microscopy. Initially, leaf specimens were fixed in FAA solution (Yuanye Bio-Technology Co., Ltd., Shanghai, China), which comprises 5% formaldehyde, 6% acetic acid, and 45% ethanol, for one hour. This was followed by a sequential dehydration process using ethanol concentrations of 70%, 85%, 95%, and ultimately 100%, with each stage lasting one hour. Subsequently, the samples underwent a solvent transition from 100% ethanol to 100% xylene to prepare them for embedding. The prepared samples were then sectioned into 0.2 μm slices using a microtome (JY202A, Beijing, China). These sections were subjected to Carnoy’s fixative, which was precooled and applied under negative pressure for 30 min, followed by further fixation at low temperatures ranging from 1 to 3 h. After fixation, the sections were rehydrated in 75% ethanol and then embedded. Dewaxing of the sections preceded the staining process, wherein they were immersed in safranin staining solution for 1–2 h and then washed in distilled water to remove any excess dye. A graduated alcohol series—75%, 85%, and 95%—was employed for decolorization, with each stage lasting 5 s. The sections were then stained with fast green for 15 s and swiftly rinsed with 95% ethanol. Elution was achieved with two treatments of absolute ethanol, each lasting 3 min. The final preparation involved clarification with xylene and sealing with neutral gum. The stained sections were examined under UV illumination via fluorescence microscopy to observe the autofluorescence characteristic of lignin, allowing for precise visualization of its distribution within the leaf tissue.

### 4.6. RNA Extraction and Real-Time Quantitative PCR

According to the manufacturer’s instructions, RNA was extracted from 200 mg of frozen tissue using the TransZol Up Plus RNA kit (Lot # Q41020, TransGen, Beijing, China) and treated with DNase. After heating and inactivation, the RNA was used for reverse transcription. The integrity and purity of the RNA were assessed using the NanoDrop 8000 spectrophotometer (Thermo Fisher Scientific Inc., Logan, UT, USA), and its concentration was quantified using the Agilent Bioanalyzer 2100 (Agilent Technologies Inc., Santa Clara, CA, USA). For reverse transcription, the EasyScript One-Step gDNA Removal and cDNA Synthesis SuperMix (Lot # P20708, TransGen, China) was employed, selecting the *Actin* gene (GenBank accession number: KC775782.1) as a reference (Ct: 22-23). To ensure the robustness and reproducibility of the findings, three independent biological replicates were performed, adhering to the protocol provided with the PerfectStart™ Green qPCR SuperMix (TransGen Biotech, Beijing, China). The quantitative real-time PCR (qRT-PCR) analyses were conducted using the ABI QuantStudio™ 6 Flex Real-Time PCR System (ABI, Carlsbad, CA, USA). The 2^−∆∆Ct^ method was adopted for normalization and calculation of relative gene expression levels. The specific primers used for the qPCR assays are detailed in Appendix A.

### 4.7. Subcellular Localization of PAL33 and Its Vector Construction for Agrobacterium-Mediated Wheat Transformation

Specific primers were designed to target the full-length CDS of the *TaPAL33* gene, with sequences 5′-ACGGGGGACTCTTGAATGGCG-GCCAACGGCAAC-3′ and 5′-GTCACCTGTAATTCACGGAGACGATTAGTCTC-3′. A 2127 bp fragment was amplified via PCR and subsequently cloned into the pMD18-T vector. In parallel, the PCR product underwent homologous recombination into the pBWA(V)HS-gfp vector (Biorun Biosciences Co., Ltd., Wuhan, China), resulting in the creation of the fusion vector Ubi-PAL33–GFP. After sequencing and validation of both the fusion and control vectors (pBWA(V)HS-gfp), these vectors were transformed into the Agrobacterium tumefaciens GV3101 strain for transient transformation experiments on tobacco leaves. The transformed leaves were then cultured on MS medium for 48 h, followed by live cell imaging utilizing an inverted confocal microscope (Zeiss LSM 780, Jena, Germany). Simultaneously, the *TaPAL33* gene was inserted into the pAHC25 vector (Miaoling Biology, Wuhan, China), which includes the Ubi promoter. This gene cassette was subsequently integrated into the pCAMBIA1301 vector (Miaoling Biology, Wuhan, China). The resulting pCAMBIA1301-PAL33 vector was sequenced and validated prior to its use in Agrobacterium-mediated transformation of wheat. To confirm the segregation of target and selectable marker genes in the transgenic plants, at least ten T0 generation plants were selected for GUS staining and resistance analysis using 150 μg/mL hygromycin (Vetec, Sigma-Aldrich, Shanghai, China). Similar analyses were also performed on T1 and T2 progeny to support further research.

### 4.8. PAL Activity Determination

This study investigated two T2 transgenic wheat lines (OE1, OE2) exhibiting the highest *TaPAL33* expression levels. Planting and growth protocols were consistent with those previously described. Both wild-type (WT) and overexpressing (OE) lines were cultivated under control conditions and subjected to high-temperature stress for 30 days. Subsequently, leaf samples were immediately harvested to evaluate PAL enzyme activity and other physiological parameters. Protein isolation and dynamic testing were conducted following the methods established by Cheng and Breen [33]. Wheat leaf samples, each weighing 0.2 g and preserved at −80 °C, were pulverized and then washed with 1 mL of ice-cold acetone (Regal Biology Technology Co., Ltd., Shanghai, China). The homogenate was incubated at −20 °C for 15 min and subsequently centrifuged at 16,000× *g* for 15 min at 4 °C. The resulting sediment was resuspended by gentle stirring in 100 mM borate buffer (Regal Biology Technology Co., Ltd., Shanghai, China) at 4 °C. After a 60 min incubation, the homogenate was centrifuged again under the same conditions to yield a clear supernatant for kinetic analysis. PAL enzyme activity was assessed by monitoring the production of trans-cinnamic acid, with absorbance readings taken at 290 nm over a period of 20 min at 37 °C [34]. The assay mixture, totaling 1 mL, consisted of 61 mM L-phenylalanine (Promega, Madison, WI, USA), 30 mM sodium borate buffer (pH 8.8), and 75 μL of the clarified supernatant. This mixture was preheated at 37 °C for 10 min prior to substrate addition. For baseline corrections, a control sample containing only the buffer (without the substrate) was pre-incubated under identical conditions. Replicates were conducted in triplicate for each testing condition. The enzymatic activity was quantified as units per milligram of protein, with one unit of PAL defined as the production of 1 μg of cinnamic acid per hour.

### 4.9. Data Analysis

Statistical analyses were conducted using SPSS V20 software (SPSS, Inc., Chicago, IL, USA). An analysis of variance (ANOVA) was performed to explore differences in gene expression across the study groups. Post hoc comparisons were carried out using Fisher’s Least Significant Difference (LSD) tests. Additionally, Duncan’s multiple range tests were employed to further discern significant differences between means, with a significance threshold set at *p* < 0.05. For the visual representation of the data, Origin 2021 software (OriginLab Corporation, Northampton, MA, USA) was utilized.

## 5. Conclusions

This study systematically analyzed the physiological and molecular responses of wheat to high-temperature stress, focusing on lignin biosynthesis. Through transcriptome analysis, we identified 9071 genes and 13 significant modules associated with high-temperature treatments, emphasizing the importance of lignin biosynthesis pathways. GO and KEGG pathway analyses highlighted the enrichment of lignin biosynthesis-related modules, particularly at the H-10DAT, H-20DAT, and H-30DAT time points. Our experiments showed a notable increase in lignin content in wheat under high temperatures, confirmed by tissue staining methods. This increase suggests an adaptive response, enhancing the plant’s defense mechanisms. The phenylalanine ammonia-lyase gene (TaPAL33) was significantly upregulated, peaking at H-30DAT, indicating its critical role in this process. Overexpression of TaPAL33 in ‘Xinchun 11’ wheat resulted in higher lignin synthesis and improved antioxidant defenses, reducing oxidative damage. These findings demonstrate that high temperatures induce the upregulation of lignin biosynthesis genes, leading to increased lignin accumulation and improved thermal tolerance in wheat. The study highlights the potential of targeting lignin biosynthesis pathways, particularly TaPAL33, to develop heat-resistant wheat varieties. This research provides a genetic foundation for enhancing wheat resilience to climate change and extreme temperatures.

## Figures and Tables

**Figure 1 plants-13-01832-f001:**
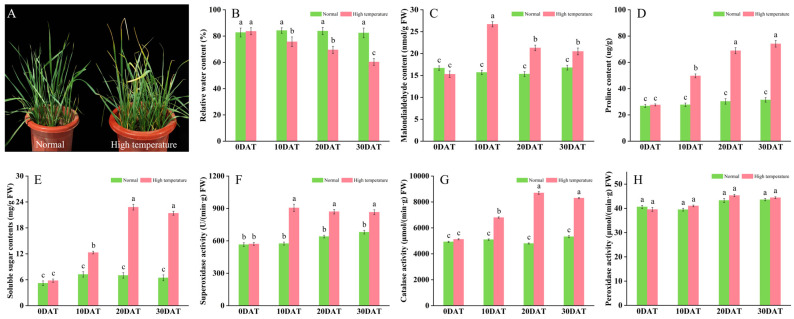
Effects of high-temperature stress on wheat plant development. (**A**) Plant morphology; (**B**) relative leaf moisture content; (**C**) malondialdehyde content (nmol/g FW); (**D**) proline content (µg/g); (**E**) soluble sugar content (mg/g FW); (**F**) superoxide dismutase activity (U/min·g FW); (**G**) catalase activity (µmol/min·g FW); and (**H**) peroxidase activity (µmol/min·g FW). Data are presented as means ± SD (*n* ≥ 5). Statistically significant differences between treatments and controls are indicated by different letters, based on Student’s unpaired two-tailed *t*-test (*p* < 0.05). The green bar chart represents normal conditions, while the red bar chart denotes high-temperature stress.

**Figure 2 plants-13-01832-f002:**
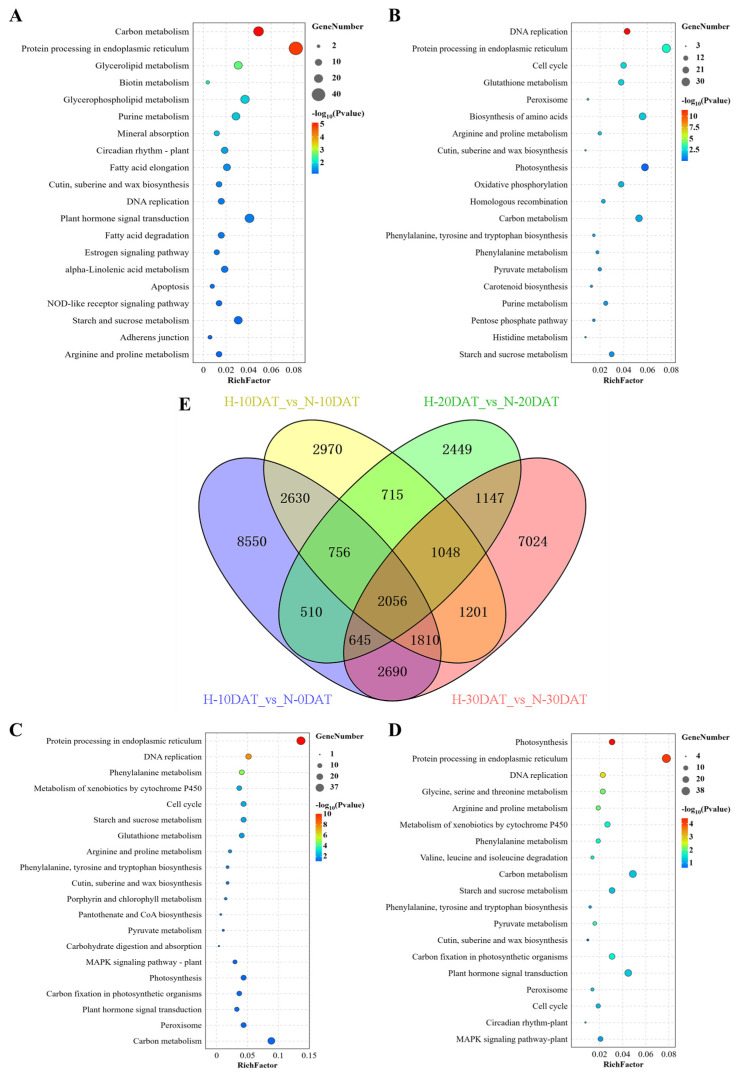
Identification and functional characterization of differentially expressed genes (DEGs) across all comparison groups. (**A**) H-10DAT_vs_N-0DAT; (**B**) H-10DAT_vs_N-10DAT; (**C**) H-20DAT_vs_N-20DAT; (**D**) H-30DAT_vs_N-30DAT; and (**E**) Venn diagrams showing DEGs among the four comparison groups (FDR < 0.05 and FC ≥ 2).

**Figure 3 plants-13-01832-f003:**
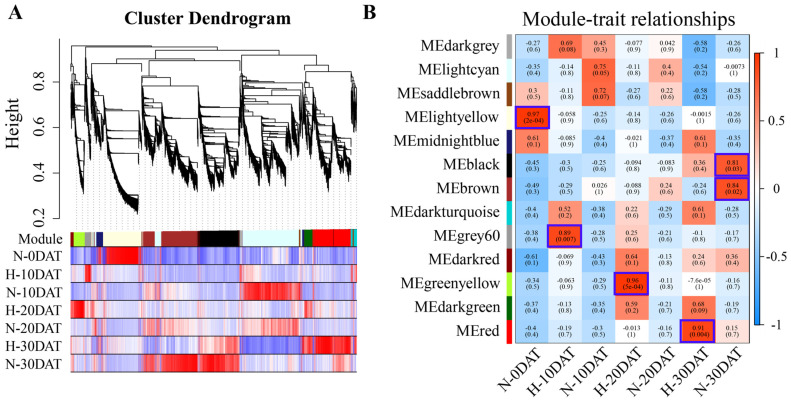
WGCNA of “XC-MU201” under various high-temperature conditions. (**A**) The hierarchical clustering tree illustrates the co-expression modules identified by WGCNA. (**B**) Module–trait relationships: Correlations between modules (left) and traits (bottom) are depicted. Positive correlations are shown in red, negative correlations in blue, and the correlation coefficients are provided. Parenthetical values indicate the corresponding *p*-values.

**Figure 4 plants-13-01832-f004:**
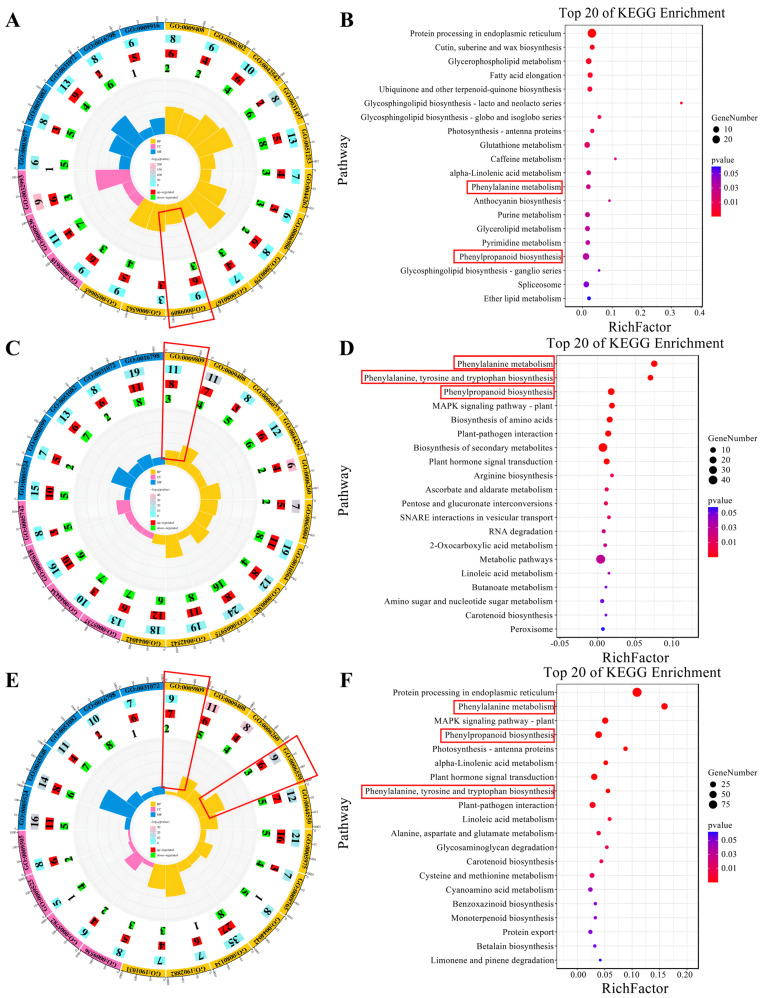
Various module genes and their annotations are associated with different high-temperature treatments. (**A**,**C**,**E**) Highlight critical GO terms for the MEgreen60, MEgreenyellow, and MEred modules, respectively. In the diagram, the outermost circle depicts the GO classification, with the size of each circle representing the number of genes it encompasses, and distinct colors indicating different categories. The second circle inward illustrates the total number of differentially expressed genes. The third and fourth circles sequentially represent the counts of upregulated and downregulated genes, respectively. At the core, the innermost circle displays the RichFactor value for each category, where each minor division on the gridline corresponds to a RichFactor increment of 0.1. (**B**,**D**,**F**) Feature important KEGG items for the MEgreen60, MEgreenyellow, and MEred modules, respectively. The horizontal axis denotes the proportion of genes, while the vertical axis specifies pathways. The color of each dot indicates the *p*-value, and the size of the dot corresponds to the number of genes involved.

**Figure 5 plants-13-01832-f005:**
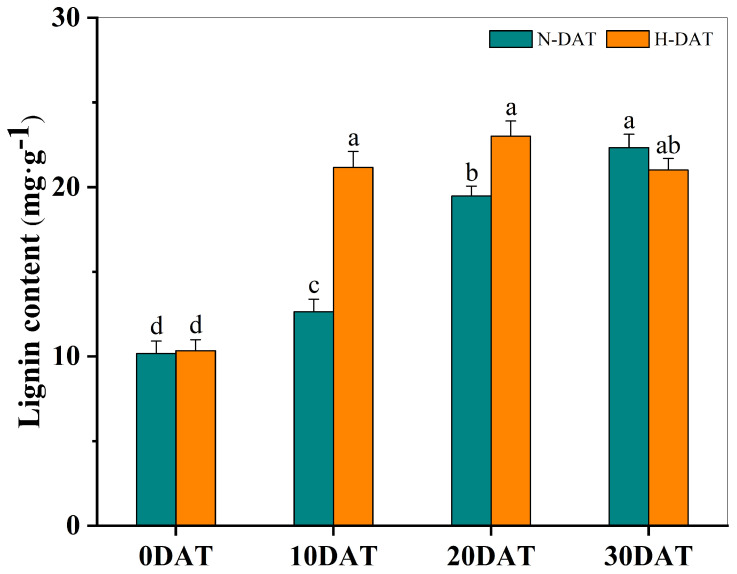
The total lignin content in XC-MU201 was analyzed under various treatment conditions, with the results graphically depicted using error bars to represent the Standard Error (SE) for three replicates (*n* = 3). The data are categorized by the duration of treatment: ‘N-DAT’ indicates days under normal conditions, and ‘H-DAT’ denotes days following high-temperature exposure. Statistical analysis revealed that differences in lignin levels, which are marked with the same letter, are not statistically significant (*p* < 0.05).

**Figure 6 plants-13-01832-f006:**
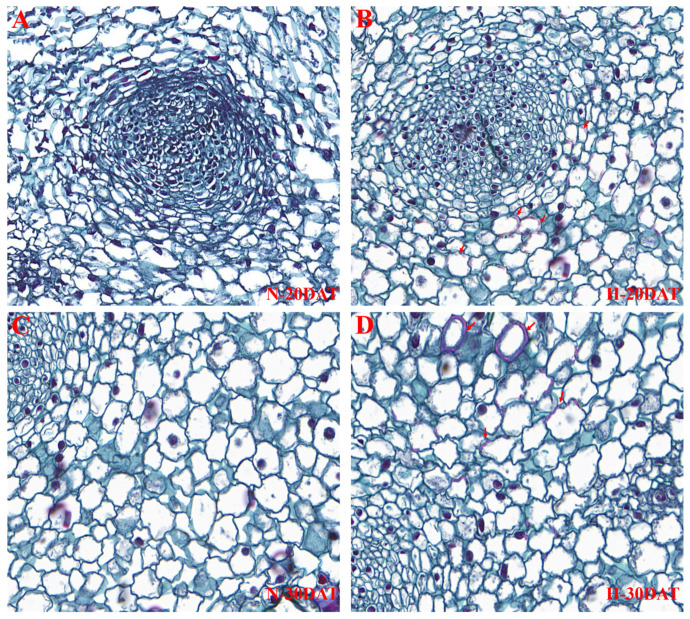
Wheat leaves from the ‘XC-MU201’ variety were subjected to four different treatments and subsequently stained with safranin. (**A**–**D**) Depict the staining outcomes for the N-20DAT, H-20DAT, N-30DAT, and H-30DAT treatments of XC-MU201, respectively, each at a magnification of 40×.

**Figure 7 plants-13-01832-f007:**
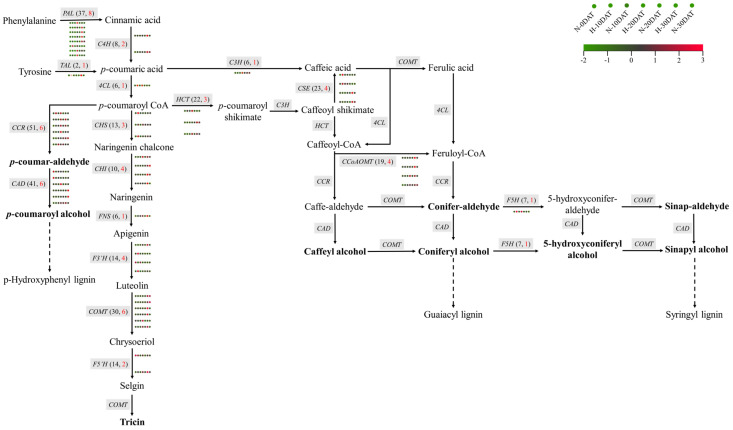
The metabolic pathways involved in the synthesis of wheat lignin monomers during exposure to high temperatures. The heatmap includes seven circles, each corresponding to a distinct treatment condition. Different genes are indicated by gray boxes. The total count of genes is shown in black text within parentheses, and the count of DEGs is highlighted in red font. The names of the lignin monomers are emphasized in bold type.

**Figure 8 plants-13-01832-f008:**
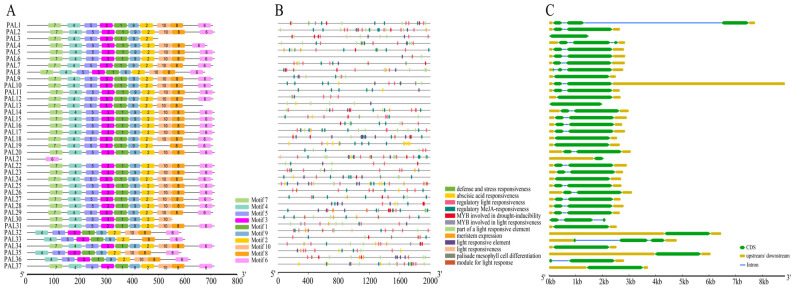
Characterization of *TaPAL* gene family members in wheat. (**A**) Conserved motif positions. The motif composition of TaPAL proteins as identified using MEME suite. Different colored boxes represent distinct motifs and their respective positions within the protein sequences. (**B**) Distribution of promoter components. This panel displays the diversity of cis-acting elements within the promoter regions of *TaPAL* genes. The variety and frequency of these elements are indicated. (**C**) Exon–intron gene structure features.

**Figure 9 plants-13-01832-f009:**
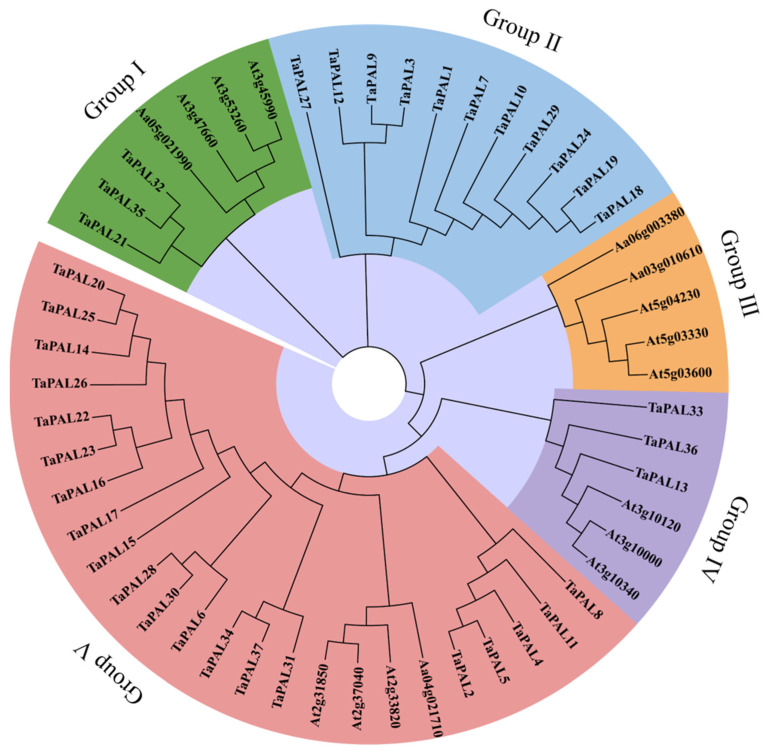
Phylogenetic analysis of PALs in wheat and Arabidopsis. This tree was constructed using the Maximum Likelihood (ML) method in MEGA7, with support values derived from 1000 replicates. Branches are color-coded to represent five distinct groups.

**Figure 10 plants-13-01832-f010:**
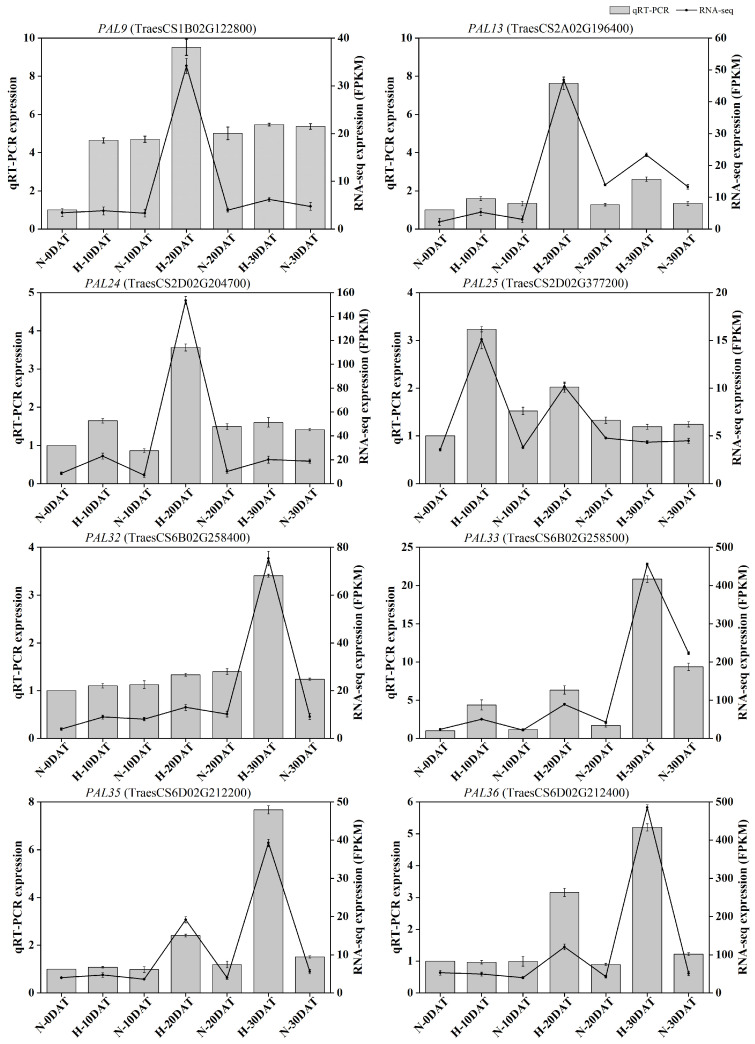
Quantitative real-time PCR analysis of eight TaPAL DEGs. The bar chart displays qRT-PCR data, and the line chart illustrates RNA-seq data. All data are presented as means ± SE.

**Figure 11 plants-13-01832-f011:**
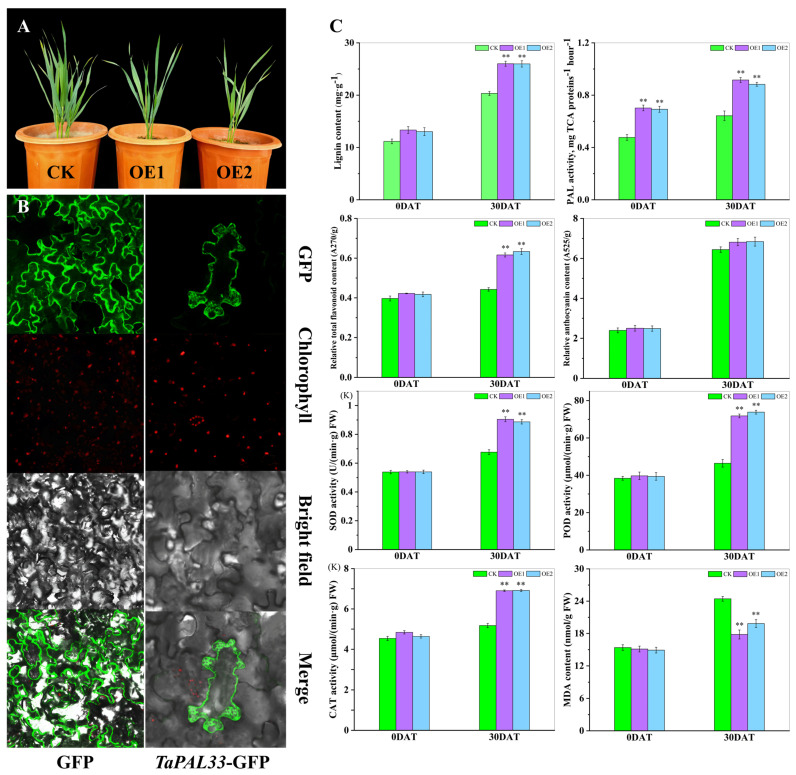
Subcellular localization of *TaPAL33* and evaluation of physiological markers in OE and WT wheat lines. (**A**) Growth dynamics of OE and WT lines. (**B**) Subcellular localization of *TaPAL33* in tobacco. For this study, the coding sequence of *TaPAL33*, which lacks a termination codon, was cloned into the pBWA(V)HS-gfp vector. This vector was subsequently introduced into four-week-old tobacco plants through Agrobacterium-mediated transformation. Localization of the TaPAL33–GFP fusion protein was investigated using a laser confocal microscope at the site of injection. GFP represents the fluorescence emitted by the empty vector protein, TaPAL33–GFP signifies the fusion protein, and chlorophyll indicates the natural fluorescence from chloroplasts. (**C**) Measurement of physiological indicators in leaves. The quantified physiological parameters of the leaves, with values expressed as the mean ± SE from three independent biological experiments. ** Highlights a statistically significant difference (*p* < 0.01) between the wild-type and transgenic wheat lines under conditions of high temperature.

## Data Availability

The raw sequencing data mentioned in this article have been archived in the Genome Sequence Archive (GSA) at the National Genomics Data Center, hosted by the China National Center for Bioinformation and the Beijing Institute of Genomics, Chinese Academy of Sciences. These data are publicly available under the accession number GSA: PRJCA026668 at https://ngdc.cncb.ac.cn/gsa (accessed on 21 January 2024).

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
