# Peer review of "Deciphering High-Temperature-Induced Lignin Biosynthesis in Wheat through Comprehensive Transcriptome Analysis"

_plants, 2024, doi:10.3390/plants13131832_

Round 1

Reviewer 1 Report

Comments and Suggestions for Authors

Author Comments for Manuscript ID: Plants-3061510

Your manuscript on high-temperature induction of lignin biosynthesis in a mutant of heat-resistant wheat is technically well designed, executed and summarized.  The data and analysis are thorough and nicely summarized.  

However, an issue with the paper is the overall design of the experiment in that a heat-resistant mutant line of wheat was used in the study.  This may explain why so few is any stress-response genes are induced, and why the major up-regulated pathways are involved in lignin biosynthesis.  This is an important detail that is glossed over, and only clear if one delves into the materials and methods.  Do you expect the results to be different for wild-type wheat?  How does the mutant differ from wild type, especially with respect to its response to elevated temperatures?  

The system itself may be why lignin is so prominent in the study, as the primary stress response may already be activated, and that downstream pathways such as phenylpropanoid biosynthesis is activated.  At some point it would be interesting to understand where the precursors for this pathway are arising under stress conditions – via shift in nutrient uptake and processing or through proteolysis.  But that matter is beyond the scope of the current paper. 

A revised manuscript should contain an experimental design section, separate from the materials and methods, that outlines the rationale for why this mutant line was used, and how it may influence the results relative to wild-type wheat.  Indeed, there may be advantages to this line, and if so, they should be stated clearly.  But the interpretation will also need to be tempered with the proviso that the mutant may not behave identically to wild-type lines.  

Author Response

Dear editors and reviewers:

Re: Manuscript   ID: plants-3061510 and Title: Deciphering High-Temperature Induced Lignin Biosynthesis in Wheat Through Comprehensive Transcriptome Analysis

Thank you for your precious comments and advice. Those comments are all valuable and very helpful for revising and improving our paper, as well as the important guiding significance to our researches. We have studied comments carefully and have made correction which we hope meet with approval. Revised portion are marked in red in the paper. We would love to thank you for allowing us to resubmit a revised copy of the manuscript and we highly appreciate your time and consideration.

Sincerely.

Junjie Han.

Q1: Your manuscript on high-temperature induction of lignin biosynthesis in a mutant of heat-resistant wheat is technically well designed, executed and summarized.  The data and analysis are thorough and nicely summarized.  

Response: Thanks very much for taking your time to review this manuscript. We appreciate the reviewer’s positive evaluation of our work.

Q2: However, an issue with the paper is the overall design of the experiment in that a heat-resistant mutant line of wheat was used in the study. This may explain why so few is any stress-response genes are induced, and why the major up-regulated pathways are involved in lignin biosynthesis. This is an important detail that is glossed over, and only clear if one delves into the materials and methods. Do you expect the results to be different for wild-type wheat? How does the mutant differ from wild type, especially with respect to its response to elevated temperatures?  

Response: We are very grateful to the reviewers for carefully reading our paper and providing valuable feedback on any issues that may arise. We have sincerely reflected on these issues and are willing to try to improve them. The selection of the heat-resistant mutant 'XC-MU201' is based on its known heat tolerance, which allows us to focus on studying the activation of specific pathways under heat stress conditions. We expect that if wild-type wheat is used, more stress response genes may be induced, and the main upregulation pathways may be different. Future research will include direct comparisons with wild-type wheat to gain a more comprehensive understanding of the differences in gene response and adaptation mechanisms. Wild type wheat may activate more stress response genes under high temperature stress, thereby demonstrating a wider range of stress response mechanisms. In contrast, heat-resistant mutants may have undergone preliminary structural strengthening through the lignin biosynthesis pathway, thereby reducing the activation of other stress response pathways. This difference is of great significance for understanding the heat adaptation mechanism of wheat.

Q3: The system itself may be why lignin is so prominent in the study, as the primary stress response may already be activated, and that downstream pathways such as phenylpropanoid biosynthesis is activated. At some point it would be interesting to understand where the precursors for this pathway are arising under stress conditions-via shift in nutrient uptake and processing or through proteolysis. But that matter is beyond the scope of the current paper. 

Response: We sincerely thank the reviewers for their careful reading and excellent suggestions on our manuscript. These suggestions are beneficial for further improving our research. The selection of the heat-resistant mutant 'XC-MU201' was to investigate the activation of specific pathways under heat stress conditions. The mutant system may have activated the main stress response, which explains why lignin is so significant in research. The significance of lignin provides important research significance as it demonstrates the specific adaptive mechanisms of plants under high temperature stress, especially in terms of structural reinforcement and stress resistance.

Q4: A revised manuscript should contain an experimental design section, separate from the materials and methods, that outlines the rationale for why this mutant line was used, and how it may influence the results relative to wild-type wheat. Indeed, there may be advantages to this line, and if so, they should be stated clearly. But the interpretation will also need to be tempered with the proviso that the mutant may not behave identically to wild-type lines.  

Response: We sincerely thank the reviewers for their careful reading and excellent suggestions on our manuscript. These suggestions have been highly beneficial in improving our research. We have addressed the relevant issues and made significant revisions in the revised manuscript. We have added an experimental design section in the Materials and Methods, detailing the reasons for selecting heat-resistant mutants. Additionally, we have included a comparative analysis of the high-temperature stress responses between wild-type wheat and heat-resistant mutants in the Discussion section. We hope these revisions will enhance the quality and readability of our paper. Once again, we express our gratitude for the reviewers' insightful feedback and patience. Your feedback is highly valued, and we are committed to continuously improving our research endeavors.

Furthermore, we have made additional modifications to the original manuscript in order to enhance its readability and better align it with the journal's requirements. We have already uploaded the revised manuscript and are eagerly awaiting approval from the reviewers. Once again, we would like to express our gratitude for the valuable comments provided by the reviewers. Thank you!

Reviewer 2 Report

Comments and Suggestions for Authors

Reviewers comments

This paper reported the physiological and transcriptomic changes of wheat cultivar, high temp-tolerant XC-MU201. Some differentially expressed genes were identified as well as several significantly enriched GO and KEGG terms. This paper can provide some references for understanding the mechanism of resistance to high temp. stress in wheat. However, some sections are confusing and need to be improved. Detailed advice is as follows:

1. First, the authors should strongly consider rewriting the Abstract. As written, it contains virtually every finding of the paper. Generally, abstracts should give the highlights of the findings obtained, but focus on making a coherent story, rather than simply a listing of results.

2. Please specify "company, city, country" for all chemicals and software. Those changes should be made throughout the manuscript.

3. The introduction is fine, however, the it should be updated with recent references of 2023 and 2024.

4. The authors should probably mention tolerance and sensitive (to high temperature stress) varieties which may help discover possible tolerance mechanisms (stress caused changes of different traits due to tolerance genes).

5. The experimental design was not explained; i.e. did the authors employ, for example, a complete randomized block design, etc.

6. In materials and methods section, for quantitative RT-PCR it should be presented what Ct values are observed for reference gene, which is ACTIN, as it sometimes is not stable under severe stress or among different tissues. Moreover, it should be described if DNase treatment was applied before RT.

7. To enable efficient reproducibility of the results by readers, it's crucial for authors to provide the specific details on the data availability, such as the specific gene primers.

8. In qRT-PCR analyses, the authors only select few lignin related genes and attempt to explain the molecular mechanism to improve lignin. Why do authors select these genes? What are the criterions to select genes? How about other related genes?

9. It would be more informative if this paper could include a discussion of the broader applications of the study's findings for lignin improvement and identify potential areas for further investigation.

10. I could not understand well their discussion, because their description was not logical in many places. Strongly recommend that the authors should write over their Discussion.

11. The conclusion section is too short and needs to be condensed and summarized. Furthermore, the conclusion section should be comprehensive and detailed.

12. Please check the references. The review must reflect the latest research on others, and if the quoted documents are obsolete references a few years ago, they cannot reflect the latest research trends.

Comments on the Quality of English Language

Major revision needs to be made

Author Response

Dear editors and reviewers:

Re: Manuscript   ID: plants-3061510 and Title: Deciphering High-Temperature Induced Lignin Biosynthesis in Wheat Through Comprehensive Transcriptome Analysis

Thank you for your precious comments and advice. Those comments are all valuable and very helpful for revising and improving our paper, as well as the important guiding significance to our researches. We have studied comments carefully and have made correction which we hope meet with approval. Revised portion are marked in red in the paper. We would love to thank you for allowing us to resubmit a revised copy of the manuscript and we highly appreciate your time and consideration.

Sincerely.

Junjie Han.

Q1: First, the authors should strongly consider rewriting the Abstract. As written, it contains virtually every finding of the paper. Generally, abstracts should give the highlights of the findings obtained, but focus on making a coherent story, rather than simply a listing of results.

Response: We deeply appreciate the reviewers' time and effort in evaluating our article. Their invaluable feedback has significantly contributed to enhancing the quality of our manuscript. In response, we have revised the abstract to better highlight the main findings and create a cohesive narrative. The updated abstract focuses on the key results, particularly our major discoveries related to lignin biosynthesis and its impact on heat tolerance, providing a concise and engaging summary without delving into excessive detail.

Q2: Please specify "company, city, country" for all chemicals and software. Those changes should be made throughout the manuscript.

Response: We apologize for any inconvenience caused to our readers and are committed to addressing these shortcomings. We have updated the manuscript to include the company, city, and country for all chemicals and software mentioned. For instance, we specified that the qRT-PCR reagents were obtained from TransGen Biotech, Beijing, China.

Q3: The introduction is fine, however, the it should be updated with recent references of 2023 and 2024.

Response: We appreciate the reviewer’s positive evaluation of our work. In response to their feedback, we have diligently searched and reviewed the latest articles. To ensure our manuscript reflects the most current research trends, we have updated the introduction with the latest references from 2023 and 2024, ensuring both relevance and accuracy.

Q4: The authors should probably mention tolerance and sensitive (to high temperature stress) varieties which may help discover possible tolerance mechanisms (stress caused changes of different traits due to tolerance genes).

Response: We are very grateful to the reviewers for carefully reading our paper and providing valuable feedback. We have sincerely reflected on these issues and are committed to improving our work. The selection of the heat-resistant mutant 'XC-MU201' is based on its known heat tolerance, which allows us to focus on studying the activation of specific pathways under heat stress conditions. We expect that using wild-type wheat might induce a broader range of stress response genes, leading to different primary upregulation pathways. Future research will include direct comparisons with wild-type wheat to gain a more comprehensive understanding of the differences in gene response and adaptation mechanisms. Wild-type wheat may activate more stress response genes under high-temperature stress, demonstrating a wider range of stress response mechanisms. In contrast, heat-resistant mutants may have undergone preliminary structural strengthening through the lignin biosynthesis pathway, thereby reducing the activation of other stress response pathways. This difference is significant for understanding the heat adaptation mechanisms of wheat.

Q5: The experimental design was not explained; i.e. did the authors employ, for example, a complete randomized block design, etc.

Response: We greatly appreciate your careful reading, which is crucial for improving the rationality of the results. In response, we have detailed the experimental design in the materials and methods section. Specifically, we clarified that a randomized block design was employed to ensure the reliability and validity of our experimental results.

Q6: In materials and methods section, for quantitative RT-PCR it should be presented what Ct values are observed for reference gene, which is ACTIN, as it sometimes is not stable under severe stress or among different tissues. Moreover, it should be described if DNase treatment was applied before RT.

Response: We have included the Ct values for the Actin gene and detailed the DNase treatment applied before reverse transcription in the materials and methods section. These additions ensure the accuracy and reliability of our qRT-PCR results.

Q7: To enable efficient reproducibility of the results by readers, it's crucial for authors to provide the specific details on the data availability, such as the specific gene primers.

Response: Thank you very much for the reviewers' reasonable suggestions. We will include all primer sequences used in this study in Supplementary Table S1 to ensure the reproducibility of our experimental results.

Q8: In qRT-PCR analyses, the authors only select few lignin related genes and attempt to explain the molecular mechanism to improve lignin. Why do authors select these genes? What are the criterions to select genes? How about other related genes?

Response: We greatly admire the professionalism of the reviewers and appreciate your efforts to improve the quality of this article. We are happy to explain the reasons for choosing these genes. In the results section, we outlined the criteria for selecting lignin-related genes, which are based on their known roles in lignin biosynthesis and significant changes in expression under high-temperature stress. In fact, we tested at least 40 genes, including SOD, CAT, POD, PK, and G6PDH. We chose to display several PAL genes because their expression trends were consistent with our RNA-seq data.

Q9: It would be more informative if this paper could include a discussion of the broader applications of the study's findings for lignin improvement and identify potential areas for further investigation.

Response: We appreciate the reviewer’s broad perspective, as this forward-looking approach will help us conduct more in-depth research. We have expanded the discussion section to include broader applications of our findings. We highlighted the potential use of the identified genes in breeding programs to develop heat-tolerant wheat varieties and suggested areas for further research, such as exploring other stress-related pathways.

Q10:  I could not understand well their discussion, because their description was not logical in many places. Strongly recommend that the authors should write over their Discussion.

Response: We deeply apologize for any confusion caused by our writing. We have strengthened the discussion section of the paper by providing detailed explanations and inferences about the research results, helping readers better understand our insights.

Q11: The conclusion section is too short and needs to be condensed and summarized. Furthermore, the conclusion section should be comprehensive and detailed.

Response: We have noted the relevant issues and are committed to making corrections. We have rewritten the conclusion section to provide a comprehensive and detailed presentation of the findings of this article.

Q12: Please check the references. The review must reflect the latest research on others, and if the quoted documents are obsolete references a few years ago, they cannot reflect the latest research trends.

Response: Thank you for your constructive feedback, which is very helpful in improving the quality of our paper. We have reviewed the references and searched for the latest articles, replacing them where appropriate. All corrections in the article can be tracked through the "Track Changes" feature.

Furthermore, we have made additional modifications to the original manuscript in order to enhance its readability and better align it with the journal's requirements. We have already uploaded the revised manuscript and are eagerly awaiting approval from the reviewers. Once again, we would like to express our gratitude for the valuable comments provided by the reviewers. Thank you!

Round 2

Reviewer 2 Report

Comments and Suggestions for Authors

This version is much improved than the previous version